# Circulating Neoplastic-Immune Hybrid Cells Predict Metastatic Progression in Uveal Melanoma

**DOI:** 10.3390/cancers14194617

**Published:** 2022-09-23

**Authors:** Michael S. Parappilly, Yuki Chin, Riley M. Whalen, Ashley N. Anderson, Trinity S. Robinson, Luke Strgar, Thomas L. Sutton, Patrick Conley, Christopher Klocke, Summer L. Gibbs, Young Hwan Chang, Guanming Wu, Melissa H. Wong, Alison H. Skalet

**Affiliations:** 1Department of Cell, Developmental and Cancer Biology, Oregon Health & Science University, Portland, OR 97201, USA; 2Department of Biomedical Engineering, Oregon Health & Science University, Portland, OR 97201, USA; 3Department of Computational Biology, Oregon Health & Science University, Portland, OR 97239, USA; 4Department of Surgery, Oregon Health & Science University, Portland, OR 97239, USA; 5Department of Medical Informatics and Clinical Epidemiology, Oregon Health & Science University, Portland, OR 97239, USA; 6Knight Cancer Institute, Oregon Health & Science University, Portland, OR 97201, USA; 7Casey Eye Institute, Oregon Health & Science University, Portland, OR 97239, USA

**Keywords:** uveal melanoma, circulating hybrid cells, cancer biomarker, circulating tumor cells

## Abstract

**Simple Summary:**

Uveal melanoma is an aggressive cancer that begins in the eye, but often spreads to distant organs when tumor cells enter the blood. Disease spreads in nearly half of uveal melanoma patients, and is fatal. We discovered a new tumor cell population in the blood stream of cancer patients that has combined tumor cell and white blood cell features. These cells, called circulating hybrid cells, can develop into metastatic tumors. Recently, we detected circulating hybrid cells in the blood of patients with uveal melanoma. In this study we determine that the number of circulating hybrid cells in the blood at time of initial treatment of uveal melanomas predicts future development of disease spread. Circulating hybrid cells have promise as a non-invasive and repeatable “liquid biopsy” for uveal melanoma patients. Therefore, study of this newly discovered cancer cell will improve understanding of the biology of disease progression in uveal melanoma.

**Abstract:**

Background: Uveal melanoma is an aggressive cancer with high metastatic risk. Recently, we identified a circulating cancer cell population that co-expresses neoplastic and leukocyte antigens, termed circulating hybrid cells (CHCs). In other cancers, CHCs are more numerous and better predict oncologic outcomes compared to circulating tumor cells (CTCs). We sought to investigate the potential of CHCs as a prognostic biomarker in uveal melanoma. Methods: We isolated peripheral blood monocular cells from uveal melanoma patients at the time of primary treatment and used antibodies against leukocyte and melanoma markers to identify and enumerate CHCs and CTCs by immunocytochemistry. Results: Using a multi-marker approach to capture the heterogeneous disseminated tumor cell population, detection of CHCs was highly sensitive in uveal melanoma patients regardless of disease stage. CHCs were detected in 100% of stage I-III uveal melanoma patients (entire cohort, *n* = 68), whereas CTCs were detected in 58.8% of patients. CHCs were detected at levels statically higher than CTCs across all stages (*p* = 0.05). Moreover, CHC levels, but not CTCs, predicted 3 year progression-free survival (*p* < 0.03) and overall survival (*p* < 0.04). Conclusion: CHCs are a novel and promising prognostic biomarker in uveal melanoma.

## 1. Introduction

Uveal melanoma (UM) is an aggressive intraocular cancer with high risk for distant metastasis driven by hematogenous dissemination of tumor cells [1]. Primary treatment of the eye is highly successful, but has no clear impact on survival [2]. This is likely due to the presence of micrometastatic disease at the time of initial diagnosis. Up to 50% of patients progress to metastatic disease over 15 years, with the most aggressive tumors often leading to metastases within 3–5 years [1]. While there is currently no curative therapy, the FDA recently approved the first uveal-melanoma-specific systemic therapy, and numerous clinical trials are underway including adjuvant treatment trials [3,4]. Increasingly, there is clinical utility in identifying patients with high risk for progression. Molecular prognostic testing can be used to risk stratify patients but requires tumor tissue obtained via fine needle biopsy, which may not be feasible for all tumors, carries risk to the patient, and is not repeatable [1,5]. Therefore, there is great interest in a non-invasive prognostic biomarker for this disease [6,7]. Circulating tumor cells (CTCs) have been investigated; however, studies examining traditional CTCs have yielded mixed, and mostly disappointing, results in part due to their rarity [6,7]. 

We identified a novel circulating tumor cell population in human cancer patients, including those with uveal melanoma, that outnumber conventionally defined CTCs [8,9]. These cells, termed circulating hybrid cells (CHCs), harbor characteristics of both neoplastic cells and macrophages [8], and thus they are identified by co-expression of tumor and leukocyte proteins. Based upon in vitro culture assays, murine models of tumorigenesis, and human patient samples, we demonstrated that tumor-macrophage hybrids retained genotypic and functional phenotypic properties of both parent cells [8,10]. Further, hybrid cells were detected within both primary and metastatic tumors, as well as in the peripheral blood [8]. CHCs harbored protein and mutational profiles that aligned with primary tumors, highlighting their direct relationship with the primary tumor [9]. Our published data supports a central role for tumor-immune hybrid cells in metastatic progression of disease, with enhanced motility, invasiveness and growth at metastatic sites [8,9]. In pancreatic ductal adenocarcinoma, CHC levels in blood correlate with AJCC staging and high levels of CHCs predict poor survival [8]. 

CHCs represent a novel tumor cell population that we recently identified in uveal melanoma patients, but have not yet extensively evaluated [9]. In the current study, we sought to evaluate CHCs as a prognostic biomarker for uveal melanoma with potential to predict metastatic disease progression. To do this, we analyzed peripheral blood specimens from patients undergoing primary treatment for stage I-III uveal melanoma. Isolated peripheral blood mononuclear cells (PBMCs) were evaluated for CHC and CTCs based upon protein expression using immunofluorescence detection. While we initially used gp100 to identify circulating tumor cell populations in our patients, we found that the use of multiple uveal melanoma markers better captured the heterogeneous population of disseminated melanoma cells present in the peripheral blood in uveal melanoma patients. This is consistent with work demonstrating downregulation of canonical genes of the melanocyte differentiation program including MITF, TRPM1, TYR and DCT in patients with high risk tumors with BAP1 mutations [3,11,12]. Differential gene expression between low risk (class 1) and high risk (class 2) tumors is the basis for molecular prognostic testing via tissue biopsy in this cancer and these differentially expressed genes impact tumor phenotype [13,14]. Thus, it follows that disseminated neoplastic cells would harbor protein expression patterns that aligned with corresponding uveal melanomas, and that they would likely be a heterogeneous population with differential expression dependent on classification of the primary tumor. 

Using both gp100 (a melanocytic marker commonly used in identification of CTCs) [14] and the cell surface serotonin receptor, 5-hydroxytryptamine receptor 2B (HTR2B), which is highly upregulated in high risk uveal melanoma tumors [15,16], we identified CHCs (gp100^+^, HTR2B^+^, CD45^+^) and CTCs (gp100^+^, HTR2B^+^, CD45^-^). We determined that CHCs significantly outnumbered CTCs in uveal melanoma patients regardless of disease stage. Progression-free survival (*p* < 0.03) and overall survival (*p* < 0.04) was significantly worse for uveal melanoma patients with high levels of CHCs in the blood at the time of primary treatment. Based upon our findings, CHCs have potential as a non-invasive biomarker in uveal melanoma. 

## 2. Materials and Methods

### 2.1. Human Specimens

All human peripheral blood and tissue samples were collected and analyzed under approved protocols in accordance with the ethical requirements and regulations of the Oregon Health & Science University institutional review board. Informed consent was obtained from all subjects. Peripheral blood was obtained from cancer patients at OHSU with uveal melanoma (*n* = 68) and healthy subjects (*n* = 18; Table 1). Primary uveal melanoma patients were risk-stratified based upon AJCC stage and, when available, gene expression profile (GEP) and Preferentially Expressed Antigen in Melanoma (PRAME) status using the commercially available DecisionDX-UM test (Castle Biosciences, Inc., Phoenix, AZ, USA) [17]. Some patients declined fine needle biopsy for GEP testing so this information was not available. Patients were followed longitudinally for clinical outcomes including local recurrence, development of distant metastases, and death.

### 2.2. Human Peripheral Blood Preparation

Patient peripheral blood was collected in heparinized vacutainer tubes (BD Biosciences, Franklin Lakes, NJ, USA) and diluted 1:2 with phosphate-buffered saline (PBS; 1.37 M NaCl, 27 mM KCl, 0.1 M Sodium phosphate di basic, 18 mM KH2PO4, pH 7.4) solution. Peripheral blood mononuclear cells (PBMCs) were isolated using density centrifugation with Ficoll-Paque PLUS (GE Healthcare, Chicago, IL, USA) as previously published [9]. Ficoll (12 mL) was layered underneath diluted blood, then centrifuged for 20 min at 800× g with no brake. Isolated PBMCs were resuspended in buffer, then adhered to poly-D-lysine (1 mg/mL) coated slides (Millipore, Burlington, MA, USA; Fisher Scientific, Waltham, MA, USA) at 37 °C for 15 min. Cells were fixed with 4% PFA for 5 min, permeabilized with 0.5% Triton-X (Fisher Scientific, BP151-100) for 10 min, and fixed again with 4% PFA for 10 min.

### 2.3. In Situ Detection and Quantification of Circulating Hybrid Cells and Circulating Tumor Cells from Human Peripheral Blood

Prepared patient PBMCs were incubated in blocking buffer (2.5 M CaCl_2_, 1% Triton-x-100, 1% bovine serum albumin (BSA) in PBS) for 30 min, and stained with a combination of antibodies against uveal melanoma antigens gp100, tyrosinase, and/or HTR2B, a basophil antigen (CD203c), a T-regulatory cell antigen (CD25), and the pan-leukocyte antigen CD45 (Appendix A), then counterstained with DAPI. Cells were imaged using a ZEISS AxioObserver.Z1 light microscope, digitally scanned with a ZEISS AxioScanner. Z1, and analyzed using ZEISS ZEN blue software version 3.5 (Carl ZEISS AG, Oberkochen, Germany). Semi-manual quantification of CTCs and CHCs was performed as previously described [9] for randomly selected areas in a blinded fashion. Fluorescence intensity thresholds for each antibody was set based on histograms of unstained cells on a discrete area of the same slide. Cells with DAPI nuclear staining were evaluated for CD45, CD25, CD203c, and uveal melanoma protein expression. We enumerated ~50,000–75,000 cells per patient, and normalized reported values to 50,000 nuclei. CTCs were defined as uveal melanoma marker-positive (HTR2B, tyrosinase, and/or gp100) and CD45-negative. CHCs were identified as uveal melanoma marker-positive (HTR2B, tyrosinase, and/or gp100), CD45-positive, CD25-negative, and CD203c-negative. Basophils and T-regulatory cells expressed CD45 and CD203c or CD25, respectively. Total nuclei within each quantified region were digitally enumerated with a semi-automated analysis program using ZEISS ZEN blue software version 3.5.

### 2.4. Analyses of Uveal Melanomas

#### 2.4.1. Immunohistochemical and Histochemical Analyses of Globes

Formalin-fixed paraffin-embedded (FFPE) tissue sections (5 µm) from enucleated globes were deparaffinized with xylene and rehydrated with graded ethanol baths. Tissue was bleached for melanin removal in 10% H_2_O_2_ for 20 min at 65 °C, then subjected to antigen retrieval. Briefly, tissue was incubated in citrate buffer (Sigma-Aldrich, St. Louis, MO, USA, pH 6) for 30 min at 100 °C, Tris-HCL buffer (Invitrogen, Waltham, MA, USA, pH 8) for 10 min at 100 °C, then cooled to room temperature, followed by PBS washes. For protein expression analyses, tissue sections were incubated with unconjugated antibodies (Appendix A) in blocking buffer for 45 min at 20 °C or overnight at 4 °C in a humid chamber, followed by washes in 2 × SCC (Biosciences, pH 7) and detection with species-appropriate fluorescent-conjugated secondary antibodies (ThermoFisher Scientific), with directly conjugated fluorescent antibodies (gp100-AF647, tyrosinase-AF594) or with Ab-oligo antibodies [18] (tyrosinase, Appendix A). All tissues were counterstained with DAPI, and coverslips were applied with Fluoromount-G mounting media (Invitrogen). Stained tissues were scanned on the ZEISS AxioScan. Z1 (ZEISS, Germany) with a Colibri 7 light source (ZEISS). The exposure time was set based upon staining controls and images were acquired using Zen Blue image acquisition software version 3.5 (ZEISS). After image acquisition, tissues were subjected to standard hematoxylin and eosin staining, and brightfield images acquired. 

#### 2.4.2. Quantification of Discrete Uveal Melanoma Cell Percentages

Nuclei/cell segmentation and feature extraction were performed as previously described [19]. The nuclei segmentation was performed using deep learning-based segmentation, Mesmer [20] based on a nuclear marker (DAPI), and cell segmentation was performed using mathematical morphology operations such as dilation (25 pixels) to capture the cell membrane. We first binarized each marker based on manual thresholding, with all pixels above the set threshold considered positive. Positive pixel fractions for each cell were extracted from the biologically relevant compartment for each marker, i.e., positive pixel fractions for markers with known nuclear (cytoplasmic) localization were extracted from nuclear (cytoplasmic) segmentation masks. Cytoplasmic segmentation masks were computed by subtracting nuclear segmentation masks from full cell body segmentation masks. We classified positive cells with a positive pixel fraction above the set threshold (0.05) and counted positive cell numbers across patient samples.

### 2.5. Bioinformatics of scRNA-Seq Dataset

#### Differential Gene Expression Analysis, Class 2 vs. Class 1 Uveal Melanoma Tumors

scRNA-sequencing data published by Durante et al. [16] was used to evaluate the differentially expressed genes between class 2 and class 1 uveal melanoma tumor cells. We downloaded the original raw data from GEO [21], and loaded and normalized data using Seurat [22] with in-house developed R scripts. To perform differential gene expression analysis, we used the FindMarkers function with logfc.threshold equal to 0 after selecting class 1 tumor cells from two primary class 1 tumor samples, UMM062 and UMM065 and selecting class 2 tumor cells from 6 primary class 2 tumor samples, UMM059, UMM061, UMM063, UMM064, UMM066, and UMM069. We randomly sampled class 2 cells so that two groups have the same number of cells (12,513 cells each). Genes with an adjusted *p* value > 0.05 and a log fold change >0.5 or <−0.5 were considered significant in differential expression between class 2 and class 1 uveal melanoma tumor cells. Volcano plots were generated using the EnhancedVolcano R package version 1.14.0 [23] in Rstudio (Boston, MA, USA).

### 2.6. Statistical Analysis

#### 2.6.1. Enumeration of CHCs and CTCs

Fisher’s exact test or Fisher’s one-tailed *t*-test was performed for comparisons of CHC and CTC proportions for each disease stage and healthy control, with CHC and CTC levels normalized to 50,000 nuclei. Fischer’s exact test was chosen as the primary test because values of zero do not change with normalization. All analyses were performed using SPSS Version 28 (IBM Corp., Armonk, NY, USA).

#### 2.6.2. Survival Analyses

Descriptive statistics were tabulated as median and interquartile range (IQR) for continuous variables and number (%) for categorical variables. Normalized CHC and CTC levels from patients and healthy subjects were compared across disease state using the Kruskal–Wallis test. Progression-free and overall survival were evaluated by Kaplan–Meier analysis using the log-rank test between groups. In order to evaluate the robustness of the correlation of CHC and CTC levels with outcomes of interest, receiver operating characteristic (ROC) curves were analyzed and multiple cutoff points were tested for significance, with the most significant threshold displayed. All patients not experiencing the event of interest were censored at the date of last surveillance imaging follow-up (for progression-free survival) or in-person follow-up (for overall survival). All tests were two-sided, and all statistical operations were performed using SPSS Version 28 (IBM Corp., Armonk, NY, USA).

## 3. Results

### 3.1. Proteins in the Melanogenesis Pathway Identify Circulating Hybrid Cells in Patients with Uveal Melanoma

The melanosomal matrix protein PMEL17/gp100 [24], is a membrane-bound protein that functions in early melanogenesis, in melanosome biogenesis and in melanin polymerization and may be clinically used to identify melanoma by histopathology or cytology [1,7]. Based upon the use of gp100 to identify conventionally defined CTCs [24] we employed antibodies against gp100 and tyrosinase, a second protein in the pathway [1,7] to detect CHCs in patients with uveal melanoma. Whole blood was collected from uveal melanoma patients immediately prior to their primary treatment. Peripheral blood mononuclear cells (PBMCs) were isolated and processed onto glass slides, then stained with antibodies against gp100 or tyrosinase (TYR) and CD45. As we previously detected [9], CHCs (gp100^+^/CD45^+^, or TYR^+^/CD45^+^) were readily identified in all uveal melanoma peripheral blood specimens, while CTCs (gp100^+^/CD45^−^ or TYR^+^/CD45^−^) were found at low levels or absent in all patients, (Figure 1A,B and Appendix A). 

To determine if presence of CHCs correlated with disease burden, we compared enumeration of CHCs and CTCs in patients presenting with uveal melanoma across disease stage (stage I, *n* = 19; stage II, *n* = 31; stage III, *n* = 13). Clinical staging of uveal melanoma patients by American Joint Committee on Cancer (AJCC) staging carries prognostic value, where stage I patients have a lower risk for metastatic spread of their disease than stage II or III patients [25,26]. Interestingly, the levels of CHCs or CTCs did not correlate with advanced stage, like they have in other cancer organ sites, such as pancreatic ductal adenocarcinoma (PDAC) [8] (Figure 1B). While AJCC stage correlated with progression-free survival as anticipated based upon published studies (Appendix A) [25,26], the levels of CHCs and CTCs defined by gp100 expression showed no correlation with progression-free survival or overall survival (Figure 1C,D). This result suggested that either CHCs did not have prognostic relevance in uveal melanoma, or that a population of interest was not being captured with gp100 staining. This latter hypothesis supported the observation that some genes and proteins, such as those involved in the melanocytic pathway, could be downregulated in more aggressive tumors and late stage disease [11,12,15,16,27]. Therefore, reliance on a single protein for biomarker detection could lead to under-detection of CHCs or CTCs if protein expression is heterogeneous across the tumor.

### 3.2. Variable RNA Expression of Key Uveal Melanoma Identifiers across the Disease Spectrum

Previous reports point to significant differential gene expression in primary uveal melanoma tumors that metastasize as compared to tumors that do not metastasize [12,16]. In order to improve detection of CHCs and CTCs in peripheral blood, we evaluated high-expressing genes associated with metastatic progression for their appropriateness to be included in a detection assay (Appendix A). While CTCs do not express peripheral blood cell antigens, CHCs, on the other hand can be generated by cell fusion between neoplastic cells and immune cells, and thus express CD45, a pan-leukocyte antigen [8,9,10]. Due to the possibility of co-expression of immune cell antigens on CHCs, we evaluated the gene list for immune cell expression and found that all top genes were co-expressed on subsets of peripheral blood mononuclear cells (Appendix A). However, the serotonin receptor, 5-hydroxytryptamine receptor 2B (HTR2B), displayed only low expression on subsets of basophils and T-regulatory cells and functions independently from the melanogenesis pathway. Therefore, we considered HTR2B as a potential candidate. To validate the use of HTR2B to identify disseminated uveal melanoma CHCs and CTCs in peripheral blood, we evaluated the single-cell RNA-seq data set generated by Durante, et.al. [16], which compared low risk (class 1) and high risk (class 2) primary uveal melanomas. Analyses of differentially expressed genes identified 1882 highly upregulated genes between these two classes, including HTR2B (Figure 2A). We then examined the correlation of altered HTR2B expression on progression-free survival in primary uveal melanomas from the TCGA database [28] and found that patients with increased HTR2B expression correlated with development of metastasis (*n* = 79, *p* = 0.005; Figure 2B). To validate that the differential gene expression matched changes in protein expression, we stained enucleated globes containing uveal melanoma tumors from *n* = 2 low risk and *n* = 2 high risk tumors for HTR2B, gp100 and TYR expression (Figure 2C, Appendix A). As anticipated, both gp100 and TYR protein expression was higher in low risk tumors and had diminished expression in high risk tumors (Figure 2D). Conversely, HTR2B expression was increased in high risk tumors. Overall, this data indicated a level of heterogeneity across tumors and across disease spectrum that could underlie challenges with under-detection of disseminated tumor cells in the blood, thus complicating their use as a prognostic biomarker.

### 3.3. Multi-Protein Antibody Cocktail Improves the Detection of CHCs in Uveal Melanoma Patients and Reflects Aggressive Disease Status

To determine if the addition of antibodies against HTR2B could increase the sensitivity of CHC and CTC detection, we first validated the use of the HTR2B antibody in our assay by demonstrating that co-staining with antibodies to CD203c and CD25, basophil and T-regulatory cell antigens, permits exclusion of these peripheral blood cell populations from our analyses (Appendix A). We then re-analyzed a subset of uveal melanoma patients across all stages of disease using a cocktail of antibodies (gp100, HTR2B, CD45, CD25, CD203c) for CHC and CTC detection. Whole blood was collected from patients with stage I-III uveal melanoma prior to their treatment (stage I, *n* = 9; stage II, *n* = 19; stage III, *n* = 11). Isolated peripheral blood mononuclear cells (PBMCs) were processed on glass slides, then stained with antibodies. CHCs (gp100^+^,HTR2B^+^/CD45^+^/ CD25^−^/CD203c^−^) were readily identified in all uveal melanoma patients’ peripheral blood specimens, while CTCs (gp100^+^,HTR2B^+^/CD45^−^/CD25^−^/CD203c^−^) were found in only a subset (Figure 3A,B). The levels of CHCs were significantly different in stage II and III patients compared to healthy controls using the Kruskal–Wallis analyses (*p* ≤ 0.007 and *p* ≤ 0.001, respectively), however we did not observe significant differences between AJCC stages (Figure 3B). Further, evaluation of protein expression heterogeneity in CHCs from a subset of patients with different stages of disease was readily apparent (Figure 3C) with specific variation in CHC protein expression from a class 2 patient (Figure 3D), highlighting the extent of tumor cell heterogeneity across the disease spectrum. This emphasizes the need to consider biologic variation to improve biomarker detection.

### 3.4. High Levels of CHCs Predict Survival Outcomes in Early Stage Uveal Melanoma Patients

To determine if enhanced detection of disseminated tumor cells into peripheral blood resulted in a clinically meaningful correlation with disease progression, we evaluated the data for progression-free (PFS) and overall survival (OS) using the log-rank test between groups for Kaplan–Meier analyses. Receiver operating characteristic (ROC) curve analysis of CHCs detected using the multi-protein antibody cocktail showed an area under the curve of 0.744 for PFS and 0.696 for OS, and identified a cutoff of >8 CHCs/50,000 nuclei for both outcomes. Using this threshold, we found that high levels of CHCs prognosticated PFS (*p* = 0.03) and OS (*p* = 0.04), whereas presence of greater than 0 CTCs/50,000 nuclei did not (Figure 3E–H). These results highlight the exciting potential of CHC detection, using multiple proteins to address heterogeneity attributed to disease evolution, in providing clinically meaningful analytes for management of patients with uveal melanoma. 

## 4. Discussion

Disseminated cancer cells drive metastatic progression and mortality in most cancer types. In several cancer types CTCs, one population of disseminated cancer cells, have demonstrated promise as a biomarker useful in predicting prognosis as well as monitoring disease progression and treatment response and are now in clinical trials [17,29,30,31,32]. In uveal melanoma conventionally defined CTCs have not demonstrated clinical utility due to a variety of biological and methodological issues [6,7]. Chief among these is low sensitivity, even in the metastatic setting. Recently, our group has identified a novel population of disseminated cancer cells, circulating hybrid cells (CHCs), which share phenotypic and genotypic identity with both cancer cells and macrophages, extending preceding hypotheses and studies of tumor-derived hybrid cells [33,34,35]. CHCs are disseminated cancer cells with enhanced migratory function, increased tumorigenicity, and greater metastatic potential than CTCs [8]. In this study, we demonstrate that CHCs are present in primary uveal melanoma patients, greatly outnumber traditional CTCs, and predict progression-free survival. Our findings are important because they identify a novel prognostic biomarker in uveal melanoma. Moreover, we describe a previously unexplored cell type, which may provide insight into disease biology of uveal melanoma, a rare but aggressive cancer type that leads to metastatic death in up to 50% of patients. Improved understanding of uveal melanoma biology and the metastatic process is critical to increased survival. 

While a number of other biomarkers have been explored in uveal melanoma, including circulating tumor DNA and microRNA, CTCs have been of particular interest [6,7]. Multiple groups have investigated CTCs as a biomarker in uveal melanoma using a variety of direct and indirect methods including RT-PCR, size filtration and immunodetection [36,37,38,39,40,41,42,43,44,45,46,47,48,49,50,51]. The rate of CTC detection in both primary and metastatic uveal melanoma is highly variable in the literature, regardless of identification method [6,7], although sensitivity in many studies is low, even in the metastatic setting. In our study, CHCs outperformed CTCs in a number of regards, including sensitivity. The rate of CHC detection was far greater than CTCs in our cohort (100% versus 58.8%). CHCs were detected at levels statistically greater than CTCs across all stages (*p* = 0.05). This permits stratification of patients by CHCs levels rather than by simply the presence or absence of CTCs as previous studies have done. This distinction may prove particularly meaningful when monitoring patients longitudinally with a goal of detecting early metastatic progression or monitoring treatment response as fluctuations in CHC levels may be detectable under these circumstances. While CTC detection was significantly lower than CHC detection in our study, utilizing two uveal melanoma markers to detect CTCs across the clinical disease spectrum likely contributes to our relatively high detection level of CTCs as compared with many previous studies.

Importantly, CHCs outperformed CTCs in predicting progression-free and overall survival in uveal melanoma patients. Uveal melanoma is associated with high risk for distant metastases, often after a latency period of months to years. Undetectable micrometastases present at the time of initial diagnosis likely drive this process, and thus successful primary treatment of the intraocular tumor does not significantly reduce risk for distant metastasis [1]. As tissue biopsy of intraocular tumors is technically challenging, associated with risk to vision, and cannot be repeated to monitor disease status over time, there is considerable interest in the development of a biomarker to assist in prognostication and disease monitoring for uveal melanoma. CHCs are a promising new biomarker in this disease. In our study, CHC detection from a single blood sample obtained when patients presented for surgery for the treatment of their primary uveal melanoma predicted progression to metastasis (*p* < 0.03), as well as their overall survival at 36 months (*p* < 0.04). Detection of CTCs was not associated with increased risk for metastatic progression or survival in our cohort. The majority of patients in the >8 CHCs/50,000 nuclei cohort (92%) have clinically progressed to metastatic disease within 3 years of primary treatment. 

While data have been inconsistent, evaluation of the very rare CTC population in uveal melanoma has suggested a possible link between presence of CTCs and survival outcomes. Likely due in part to issues with CTC detection and sensitivity, the data demonstrating a link between CTCs and survival outcomes has been mixed, with some groups reporting a correlation between presence of CTCs and poor clinical outcome, while others show no such relationship [38,40,42,43,44,45,47,51]. As CHCs are detected at significantly higher rates than CTCs, our data indicate that combining detection of CTCs with the more abundant population of CHCs may be a more reliable approach for identifying patients at high risk for metastasis when a biopsy is not performed. While highly predictive molecular prognostic testing is available for patients in whom fine needle aspirate biopsy is feasible and desired [5,52], a blood-based biomarker permits non-invasive risk stratification which is advantageous. Repeat biopsy cannot be performed after primary treatment. Therefore, longitudinal evaluation is not possible using existing tissue biopsy-based strategies and temporal detection of CHCs would complement existing strategies for predicting clinical outcome and monitoring disease status. 

In addition to our primary findings, we determined that gp100, a marker used routinely for identification of CTCs [7] in uveal melanoma and the target for the recently FDA-approved tebentafusp for metastatic disease [4], was insufficient for identification of disseminated cancer cells in primary uveal melanoma patients with more aggressive disease. As these are the patients who stand to benefit the most from reliable non-invasive risk stratification (e.g., for adjuvant clinical trial enrollment), and early detection of metastases, the identification of HTR2B as a marker that enhances detection of CHCs in combination with gp100 to identify high risk uveal melanoma is important and timely. We also found that CHC levels increase with clinical stage based upon mean CHC levels, although there were no statistically significant differences between stages I–III. This may reflect the small numbers in our cohort as well as the wide variability in CHC levels between patients. Another possibility is that we may be missing a CHC population with discrete protein expression, based upon heterogeneous expression of uveal melanoma markers across the disease spectrum. Genetic evolution of metastatic uveal melanoma is well-documented [16,27], however phenotypic and mechanistic behaviors have not yet been attributed to many of the acquired alterations. While identifying and understanding these differences are key to identifying heterogeneous disseminated CHCs in patients with uveal melanoma, they are also key to design of effective targeted therapies. However, as the CHC levels identified based upon dual expression of gp100 and HTR2B were predictive of clinical outcome (progression to metastasis; overall survival), it is also possible that this result simply reflects the insufficiency of tumor, node, metastasis staging to predict disease progression on an individual patient basis in uveal melanoma, a cancer in which lymphatic spread is exceptionally rare. 

The finding that macrophage-melanoma hybrid cells are present in uveal melanoma patients, greatly outnumber conventionally defined CTCs in circulation, and predict survival outcomes is novel and intriguing. While it is unclear in which setting the neoplastic and immune cells are generated, our data support that this interaction may be occurring in the primary tumor microenvironment. Increasingly, the tumor immune microenvironment is appreciated as a critical factor in cancer biology, driving tumor progression and treatment response. The inflammatory subtype of primary uveal melanoma is characterized by immunosuppressive M2 macrophages and is associated with worse survival [1,53]. While direct immunosuppression of T-cell mediated killing likely plays a key role in the poor prognosis associated with immune infiltration of uveal melanomas, it is also possible that the presence of tumor-associated macrophages provide an environment rich for development of macrophage-melanoma hybrid cells in the primary tumor. These hybrid cells, poised to disseminate into circulation, could drive the metastatic process. Moreover, hybrid cell phenotypes related to autophagy [54], the glycome, and glycosyltranferase expression may provide intriguing functional insights into tumor progression. While interesting conceptually, more research is needed to determine whether hybrid cells are present in primary uveal melanomas and if so, whether there is a correlation between inflammatory tumor subtype, hybrid cell numbers within tumors, and patient survival outcomes. 

## 5. Conclusions

In this study, we identify circulating neoplastic-immune hybrid cells (CHCs) as a novel and abundant disseminated tumor cell type in primary uveal melanoma patients. For the first time, we demonstrate that CTCs are the minority of tumor cells detectable in the peripheral blood. As identified using a multi-protein immunodetection method, CHCs outnumbered CTCs in patients with primary uveal melanoma at the time of initial diagnosis and treatment. Detection of CHCs was highly sensitive in patients regardless of disease stage I-III, with CHCs detected in 100% of uveal melanoma patients compared to conventionally defined CTCs (58.8%). Further, CHCs greatly outnumbered CTCs (*p* = 0.05). Moreover, CHC levels, but not CTCs, predicted progression-free survival as well as overall survival in patients with primary uveal melanoma based upon a single blood draw at the time of initial treatment. In addition, our work highlights the heterogeneity of disseminated tumor cell populations in uveal melanoma patients, particularly those with more aggressive disease. Using multiple uveal melanoma markers, our sensitivity for CHC and CTC detection was improved, and correlations between CHC levels in the blood and clinical outcome became apparent. Further study of CHCs as a biomarker in uveal melanoma is warranted. 

Future planned studies include a longitudinal evaluation of CHC levels in early stage uveal melanoma patients to determine the impact of primary treatment on CHC levels, if any, and to evaluate serial CHC detection as a possible method for early detection of metastatic disease prior to radiologic detection on surveillance imaging. We also plan to expand our comparisons with existing tissue biopsy-based prognostic techniques such as GEP and PRAME expression as we expand our patient cohort. In addition, we intend to further evaluate CHCs in both the primary and metastatic tumor microenvironment. As macrophage infiltration is well established as a poor prognostic indicator in uveal melanoma, the presence of macrophage-melanoma hybrids in primary uveal melanomas and a potential role for these cells as drivers of metastatic progression is particularly interesting and worth further exploration. 

## Figures and Tables

**Figure 1 cancers-14-04617-f001:**
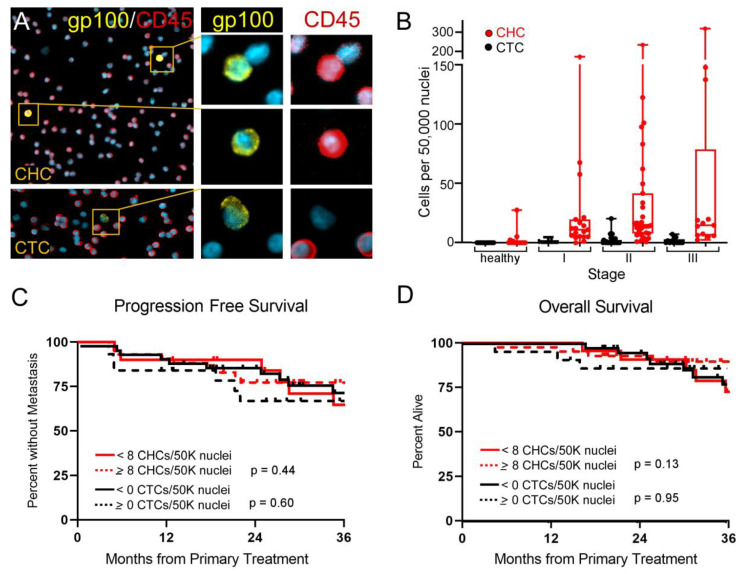
Detected gp100^+^ circulating hybrid cells (CHCs) do not correlate with uveal melanoma stage or predict survival outcomes. (**A**) Peripheral blood mononuclear cells stained with antibodies to gp100 and CD45 facilitate identification of CHCs and circulating tumor cells (CTCs). Higher magnification of boxed regions with individual antibody staining shown on right. (**B**) Enumerated CHCs and CTCs in healthy subjects and patients across AJCC disease stage. (**C**) High levels of CHCs or CTCs do not correlate with progression-free survival or (**D**) overall survival.

**Figure 2 cancers-14-04617-f002:**
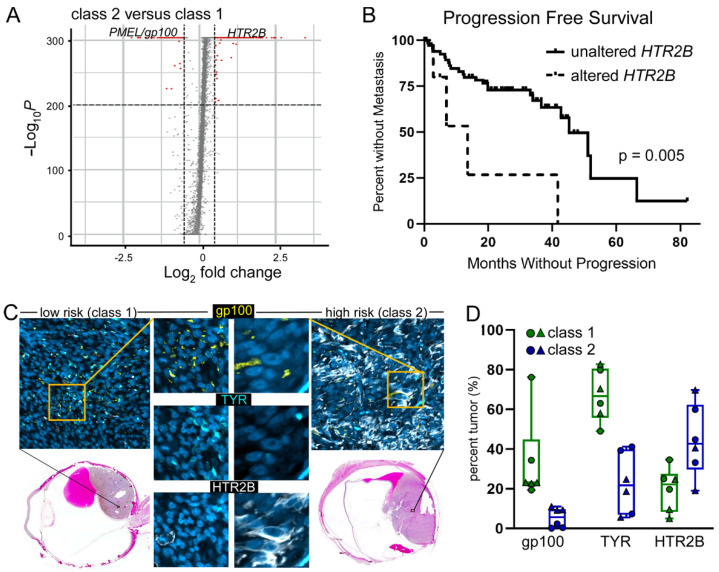
Validation of gene elevated in high risk tumors. (**A**) Volcano plot of differential gene expression between class 2 and class 1 tumors from a publicly available single cell RNA-seq dataset. (**B**) TCGA uveal melanoma dataset of *n* = 79 patients analyzed for altered expression of *HTR2B* correlated with progression free survival. (**C**) low risk (class 1) and high risk (class 2) tumors stained with antibodies to gp100 (yellow), TYR (blue) and HTR2B (white) merged in the image. Individual antibody staining shown at higher magnification of the boxed region. Hematoxylin & eosin stained image of the same region of the globe is shown. (**D**) Quantification of protein staining from three regions of interest from each tumor.

**Figure 3 cancers-14-04617-f003:**
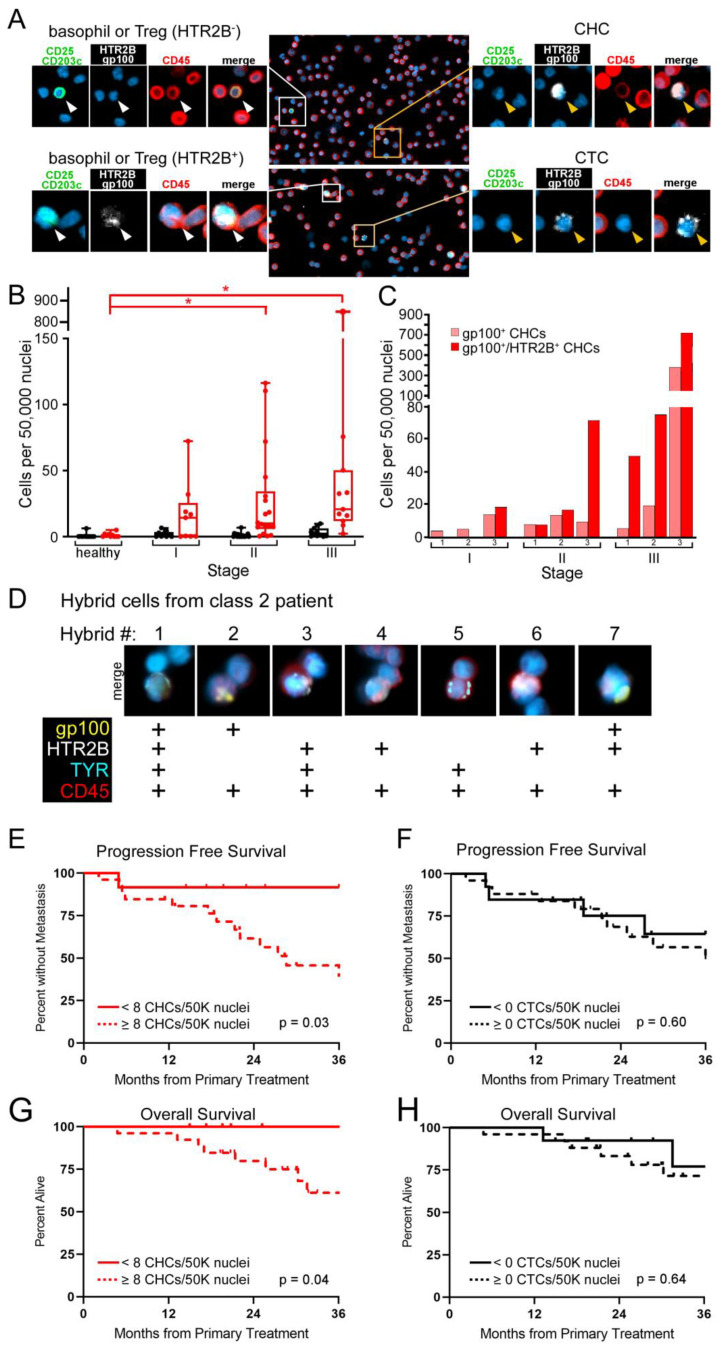
Circulating hybrid cells (CHCs) detected with an enhanced antibody cocktail correlate with progression free survival and overall survival. (**A**) Peripheral blood mononuclear cells stained with antibodies to gp100/HTR2B (white) and CD45 (red) facilitate identification of CHCs and circulating tumor cells (CTCs). CHCs are distinguished from basophils and T regulatory cells (Treg) based upon lack of expression of CD203c and CD25 (green), respectively. Higher magnification of boxed regions with individual antibody staining shown. (**B**) Enumerated CHCs and CTCs in healthy subjects and patients across stage. Asterisk indicates *p* < 0.05. (**C**) Levels of gp100^+^ CHCs and gp100^+^/HTR2B^+^ CHCs from three representative patients from each evaluated stage. (**D**) CHCs harboring heterogeneous protein expression identified in a class 2 patient. (**E**,**F**) High levels of CHCs correlate with progression free survival and overall survival, whereas there is no correlation for CTCs (**G**,**H**).

**Table 1 cancers-14-04617-t001:** Clinicopathologic Characteristics of Subjects Analyzed.

Uveal Melanoma Patients	gp100 N = 63	HTR2B Cocktail N = 38	Total Cohort N = 68
**Median age, years; (IQR)**			
	67 (58–71)	60 (56–71)	63 (56–81)
**Caucasian Race, N (%)**			
	62 (98.4)	39 (100)	67 (98.5)
**Gender, N (%)**			
F	25 (39.7)	14 (35.9)	26 (38.2)
M	38 (60.3)	25 (64.1)	42 (61.8)
**T Stage, N (%)**			
T1	23 (36.5)	10 (25.6)	23 (33.8)
T2	0 (0)	14 (35.9)	21 (30.9)
T3	33 (52.4)	8 (20.5)	16 (23.5)
T4	7 (11.1)	7 (17.9)	8 (11.8)
**AJCC Stage, N (%)**			
I	19 (30.2)	9 (23.1)	19 (27.9)
II	31 (49.2)	19 (48.7)	35 (51.5)
III	13 (20.6)	11 (28.2)	14 (20.6)
**GEP Class, N (%)**			
Class 1	23 (36.5)	12 (30.7)	25 (36.8)
Class 2	21 (33.3)	18 (46.2)	24 (35.3)
Not Assessed	19 (30.2)	9 (23.1)	19 (27.9)
**PRAME, N (%)**			
Neg	26 (41.3)	15 (38.5)	27 (39.7)
Pos	18 (28.6)	15 (38.5)	21 (30.9)
Not Assessed	19 (30.2)	9 (23.1)	20 (29.4)
**Controls**	**gp100** **N = 14**	**HTR2B cocktail** **N = 10**	**Total Cohort** **N = 18**
**Median age, years; (IQR)**			
	69 (63–75)	73 (68–78)	71 (63–76)
**Caucasian Race, N (%)**			
	13 (92.9)	10 (100)	17 (94.4)
**Gender, N (%)**			
F	10 (71.4)	8 (80.0)	13 (72.2)
M	4 (28.6)	2 (20.0)	5 (27.8)

## Data Availability

Not applicable.

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
