# Peer review of "Circulating Neoplastic-Immune Hybrid Cells Predict Metastatic Progression in Uveal Melanoma"

_cancers, 2022, doi:10.3390/cancers14194617_

Round 1

Reviewer 1 Report

This paper describes phenomena which have been partially poblished by Pawelek and Lazarova in 2009. These papers must be obligatorily described in the present paper, and the experimental schedules modified avvordingly. In particular. the analysis of glycome and glycosyltransferases performed. See: DOI:10.1111/j.1600-0625.2009.00933.x, doi: 10.1111/j.1600-0560.2009.01359.x.

Moreover, as pigmentation of the cells seems to play a key role or be a key associate with the metastatic potential of ocular melanoma, there is a striking lack of data on pigmentation of the observed hybrid cells, which must be supplemented. See also: https://doi.org/10.3390/cancers14112753

Author Response

Dear Reviewer,

Thank you for the critical review of our manuscript, “Circulating neoplastic-immune hybrid cells predict metastatic progression in uveal melanoma.” We appreciate the opportunity to address the valuable critique, and hope our manuscript is now acceptable for publication. Our response to the your comments are below.

Sincerely,

Alison H. Skalet and Melissa H. Wong

1a. This paper describes phenomena which have been partially poblished by Pawelek and Lazarova in 2009. These papers must be obligatorily described in the present paper, and the experimental schedules modified avvordingly. See: DOI:10.1111/j.1600-0625.2009.00933.x, doi: 10.1111/j.1600-0560.2009.01359.x.

We respectfully point out that our study focus is on uveal melanoma, which has significant biologic and clinical differences compared to cutaneous melanoma [1].

The reviewer highlights that in vitro-derived cutaneous melanoma-hybrid cell lines were discussed in the Lazova and Pawelek commentary in Experimental Dermatology (2009) [2], with reference to Dr. Pawelek’s earlier reviews, in Nat Rev Cancer (2008) [3] and in Advances in Cancer Research (2008) [4]. We respectfully point out that the reference to melanoma-macrophage hybrids in Lazova and Pawelek are specific to cutaneous melanoma and to in vitro-derived hybrid cell lines. Our study focus on uveal melanoma—a distinct disease from cutaneous melanoma—and a disseminated cell population in peripheral blood (versus characterization of an in vitro-derived population). These difference are such that deviating to describe and include analyses of cutaneous melanoma signatures from cell lines would dilute the impact and findings of our presented studies.

We do appreciate that cell fusion, and specifically neoplastic-immune cell fusion, is a long-standing field, and thus we now reference the intrepid investigators that have provided the foundation for our studies on page 10, Line 364-365:

“Recently, our group has identified a novel population of disseminated cancer cells, circulating hybrid cells (CHCs), which share phenotypic and genotypic identity with both cancer cells and macrophages, extending preceding hypotheses and studies of tumor-derived hybrid cells [3,5,6].”

1b. In particular. the analysis of glycome and glycosyltransferases performed.

The reviewer points out that in-depth analyses of in vitro-derived cutaneous melanoma-macrophage hybrid cells revealed autophagy phenotypes for these cells. While this is an interesting point for cutaneous melanoma and in vitro-derived hybrids (or potentially in vivo-derived hybrids located within primary tumors), our study focuses on disseminated hybrid cells collected from peripheral blood.

It is an interesting point that some of the heterogeneously expressing CHCs may have differential autophagy signatures telegraphing underlying biology of uveal melanoma. However, delving into the biology of the disseminated hybrid cells lies outside the scope of this manuscript, which is a translational biomarker study.

Phenotype analyses of the glycome and glycosyltransferase expression would require prospective collection of new patient specimens, and is not feasible, nor within the scope of this study.

Exploration of autophagy function would require an experimental murine model of uveal melanoma as well as generation of in vitro-derived hybrid cell lines in order to generate impactful and rigorous data to support any finding. Analyses of individual patient specimens that have different disease stages/states would provide too much variability to draw conclusions, but could complement future basic science investigation of hybrid cell autophagy phenotypes and function in uveal melanoma. However, as stated, this deviation in the line of experimentation is outside the scope of our study.

Despite this, we agree with the reviewer that phenotyping hybrids could reveal underlying functional biology and thus we have now included the following sentence on P12, Line 463-465:

“Moreover, hybrid cell phenotypes related to autophagy [7], the glycome, and glycosyltranferase expression may provide intriguing functional insights into tumor progression.”

  1. Moreover, as pigmentation of the cells seems to play a key role or be a key associate with the metastatic potential of ocular melanoma, there is a striking lack of data on pigmentation of the observed hybrid cells, which must be supplemented. See also: https://doi.org/10.3390/cancers14112753

While pigmentation of uveal melanoma tumors has been identified as a factor predictive for metastasis in some studies, there is currently insufficient evidence to consider pigmentation as playing a key role in the metastatic progression of uveal melanoma. Pigmentation of the tumors is not used clinically for predicting outcomes and is not included in clinical prognostication models such as the Liverpool Uveal Melanoma Prognosticator Online [8-10], nor in the American Joint Committee on Cancer Staging for uveal melanoma [11-14]. Similarly, pigmentation is not among the risk factors used to determine risk stratification in the National Comprehensive Cancer Network Uveal Melanoma Guidelines [15]. Clinically, prognostication in uveal melanoma focuses upon highly predictive molecular testing from tumor biopsy (i.e. gene expression profile testing, chromosomal analyses, and mutational analyses) as well as tumor size, anatomic location, and presence or absence of extra-ocular extension [1].

In addition to the clinical rationale for not including pigmentation as a readout for CHCs, we respectfully point out that our study identified that the melanogenesis pathway is molecularly downregulated in aggressive tumors (Figure 2, 3), and at best this downregulation results in heterogeneous phenotypes of disseminated hybrid cells within uveal melanoma patients (Figure 3C,D, Sup Figure 3B).

While there is renewed interest in the relationship between the pigmentation of both the iris and tumor in uveal melanoma biology, exploration of tumor/hybrid pigmentation is beyond the scope of this study.

Comments in response to “must be improved” checklist items:

  1. Does the introduction provide sufficient background and include all relevant references?

Our study focuses on uveal melanoma which has significant biologic and clinical differences compared to cutaneous melanoma. It does not make clear sense to reference reviews on cutaneous melanoma in the introduction. We have included a number of the recommended references that are relevant to this study in the Discussion as detailed in the response to Comment 1a. 

  1. Are all the relevant references relevant to the research?

We have reviewed our manuscript and believe that all of the citations are relevant and contributory.

  1. Is the research design appropriate? Are the conclusions supported by the results?

The research presented in this manuscript that evaluates disseminated neoplastic cells as a biomarker for aggressive uveal melanoma is appropriately designed. Our results are accurately interpreted and presented.

While the reviewer is requesting biologic phenotyping of hybrid cells, this is not the focus of our study. The manuscript reports a translational biomarker study.

References

  1. Jager, M.J.; Shields, C.L.; Cebulla, C.M.; Abdel-Rahman, M.H.; Grossniklaus, H.E.; Stern, M.H.; Carvajal, R.D.; Belfort, R.N.; Jia, R.; Shields, J.A.; et al. Uveal melanoma. Nat Rev Dis Primers 2020, 6, 24, doi:10.1038/s41572-020-0158-0.
  2. Lazova, R.; Pawelek, J.M. Why do melanomas get so dark? Experimental Dermatology 2009, 18, 934-938, doi:https://doi.org/10.1111/j.1600-0625.2009.00933.x.
  3. Pawelek, J.M.; Chakraborty, A.K. Fusion of tumour cells with bone marrow-derived cells: a unifying explanation for metastasis. Nat Rev Cancer 2008, 8, 377-386, doi:10.1038/nrc2371.
  4. Pawelek, J.M.; Chakraborty, A.K. The cancer cell--leukocyte fusion theory of metastasis. Adv Cancer Res 2008, 101, 397-444, doi:10.1016/s0065-230x(08)00410-7.
  5. Aichel, O. Über Zellverschmelzung mit Qualitativ Abnormer Chromosomenverteilung als Ursache der Geschwulstbildung [About cell fusion with qualitatively abnormal. chromosome distribution as cause for tumor formation.] In Vorträge und Aufsätze über Entvickelungsmechanik Der Organismen; 1911; pp. 92–111.
  6. Sutton, T.L.; Patel, R.K.; Anderson, A.N.; Bowden, S.G.; Whalen, R.; Giske, N.R.; Wong, M.H. Circulating Cells with Macrophage-like Characteristics in Cancer: The Importance of Circulating Neoplastic-Immune Hybrid Cells in Cancer. Cancers (Basel) 2022, 14, doi:10.3390/cancers14163871.
  7. Giatromanolaki, A.N.; Charitoudis, G.S.; Bechrakis, N.E.; Kozobolis, V.P.; Koukourakis, M.I.; Foerster, M.H.; Sivridis, E.L. Autophagy patterns and prognosis in uveal melanomas. Mod Pathol 2011, 24, 1036-1045, doi:10.1038/modpathol.2011.63.
  8. Cunha Rola, A.; Taktak, A.; Eleuteri, A.; Kalirai, H.; Heimann, H.; Hussain, R.; Bonnett, L.J.; Hill, C.J.; Traynor, M.; Jager, M.J.; et al. Multicenter External Validation of the Liverpool Uveal Melanoma Prognosticator Online: An OOG Collaborative Study. Cancers (Basel) 2020, 12, doi:10.3390/cancers12020477.
  9. DeParis, S.W.; Taktak, A.; Eleuteri, A.; Enanoria, W.; Heimann, H.; Coupland, S.E.; Damato, B. External Validation of the Liverpool Uveal Melanoma Prognosticator Online. Investigative Ophthalmology & Visual Science 2016, 57, 6116-6122, doi:10.1167/iovs.16-19654.
  10. Available online: https://mpcetoolsforhealth.liverpool.ac.uk/LUMPONet/LUMPONet.html (accessed on
  11. International Validation of the American Joint Committee on Cancer's 7th Edition Classification of Uveal Melanoma. JAMA Ophthalmol 2015, 133, 376-383, doi:10.1001/jamaophthalmol.2014.5395.
  12. Force, T.A.O.O.T. International Validation of the American Joint Committee on Cancer’s 7th Edition Classification of Uveal Melanoma. JAMA Ophthalmology 2015, 133, 376-383, doi:10.1001/jamaophthalmol.2014.5395.
  13. Amin, M.B.; Edge, S.B.; Greene, F.L.; Byrd, D.R.; Brookland, R.K.; Washington, M.K.; Gershenwald, J.E.; Compton, C.C.; Hess, K.R.; Sullivan, D.C. AJCC Cancer Staging Manual; Springer International Publishing: 2018.
  14. Amin, M.B.; Greene, F.L.; Edge, S.B.; Compton, C.C.; Gershenwald, J.E.; Brookland, R.K.; Meyer, L.; Gress, D.M.; Byrd, D.R.; Winchester, D.P. The Eighth Edition AJCC Cancer Staging Manual: Continuing to build a bridge from a population-based to a more "personalized" approach to cancer staging. CA Cancer J Clin 2017, 67, 93-99, doi:10.3322/caac.21388.
  15. Barker, C.A.; Salama, A.K. New NCCN Guidelines for Uveal Melanoma and Treatment of Recurrent or Progressive Distant Metastatic Melanoma. J Natl Compr Canc Netw 2018, 16, 646-650, doi:10.6004/jnccn.2018.0042.

Reviewer 2 Report

In the manuscript titled “Circulating Neoplastic-Immune Hybrid Cells Predict Meta-2 static Progression in Uveal Melanoma” the authors investigated the potential of circulating hybrid cells (CHCs) as a prognostic biomarker in uveal melanoma. They isolated blood monocular cells from uveal melanoma patients at the time of primary treatment and employed a multi-marker approach, finding that the detection of CHCs was highly sensitive in uveal melanoma patients regardless of disease stage. Moreover, they mainly detected CHCs respect to circulating tumor cells (CTCs). CHC levels, but not CTCs, predicted 3 year progression-free survival and overall survival providing data to support CHCs a novel and promising prognostic biomarker in uveal melanoma. The manuscript is innovative, uses a new methodological approach and applies it to uveal melanoma. The manuscript is well written and the elements are clearly described.

This version of the manuscript is believed to be suitable for publication.

Author Response

We thank the reviewer for their positive review. We believe this study will have strong impact for patients with uveal melanoma and are excited to be pioneering this field.

Sincerely,

Alison H. Skalet and Melissa H. Wong

Round 2

Reviewer 1 Report

The Authors seriously considered my review and devoted much time to explain their point of view. I still maintain my opinion that uveal melanoma is not so different tumor than skin melanoma and the question of the priority in the publication of the hybridoma cells is very difficult to solve equivocally. Melanomas share some commot features in general. As to the question of pigmentation, the scientific point of view should primarily consider the facts, and only secondary the regulations of what to consider and what not. However, as the authors cited prof. Pawelek, which was my primary remark, I accept the new version of the manuscript.